# Displacement Identification by Computer Vision for Condition Monitoring of Rail Vehicle Bearings

**DOI:** 10.3390/s21062100

**Published:** 2021-03-17

**Authors:** Lei Lei, Dongli Song, Zhendong Liu, Xiao Xu, Zejun Zheng

**Affiliations:** 1State Key Laboratory of Traction Power, Southwest Jiaotong University, Chengdu 610036, China; leilei309@my.swjtu.edu.cn (L.L.); XuXiao@my.swjtu.edu.cn (X.X.); zhengjuner@my.swjtu.edu.cn (Z.Z.); 2Department of Engineering Mechanics, KTH Royal Institute of Technology, 10044 Stockholm, Sweden; zhendong@kth.se

**Keywords:** displacement detection, bearing system, experimental platform, computer vision, phase correlation, GLCM, condition monitoring

## Abstract

Bearings of rail vehicles bear various dynamic forces. Any fault of the bearing seriously threatens running safety. For fault diagnosis, vibration and temperature measured from the bogie and acoustic signals measured from trackside are often used. However, installing additional sensing devices on the bogie increases manufacturing cost while trackside monitoring is susceptible to ambient noise. For other application, structural displacement based on computer vision is widely applied for deflection measurement and damage identification of bridges. This article proposes to monitor the health condition of the rail vehicle bearings by detecting the displacement of bolts on the end cap of the bearing box. This study is performed based on an experimental platform of bearing systems. The displacement is monitored by computer vision, which can image real-time displacement of the bolts. The health condition of bearings is reflected by the amplitude of the detected displacement by phase correlation method which is separately studied by simulation. To improve the calculation rate, the computer vision only locally focuses on three bolts rather than the whole image. The displacement amplitudes of the bearing system in the vertical direction are derived by comparing the correlations of the image’s gray-level co-occurrence matrix (GLCM). For verification, the measured displacement is checked against the measurement from laser displacement sensors, which shows that the displacement accuracy is 0.05 mm while improving calculation rate by 68%. This study also found that the displacement of the bearing system increases with the increase in rotational speed while decreasing with static load.

## 1. Introduction

Axle box bearings of rail vehicle, as a key component of railway running gear, are used to adapt the rotational movement of wheelsets into a longitudinal motion of the car body along the track. Any fault state of the axle box bearing seriously challenges the running safety of rail vehicles [1]. Different from other kinds of bearings, axle box bearings work in a harsh condition and are heavily subjected to wheel–rail contact force, dynamic vibration generated by the car body and frame, meshing excitation during the gear engagement in the power transmission of gear box, and the excited dynamic load generated by the bearing itself [2]. Therefore, it is necessary to monitor the health condition of the bearings to ensure the running safety of rail vehicles. 

At present, the condition monitoring and health diagnosis methods of bearings in railway applications are mainly categorized into two groups: on-board monitoring based on vibration or temperature, and trackside monitoring based on acoustics. The on-board monitoring requires additional detecting equipment installed on the bogie, which greatly increases the manufacturing cost. The acoustic signals measured by trackside monitoring are seriously affected by ambient noise [3,4]. In order to reduce monitoring cost and further improve monitoring accuracy and reliability, it is necessary to develop a new method for bearing monitoring. 

In contrast, there are many monitoring technologies for structural health monitoring which can potentially apply to bearing condition monitoring. Structural health monitoring is widely used for the assessment of structural performance and safety state by monitoring, analyzing, and identifying various loads and structural responses of the target structures [5,6,7]. Displacement is an important index for structural state evaluation and performance evaluation [6] because the displacement can be further converted into a corresponding physical index for structural safety assessment. Static and dynamic characteristics of the structure, such as bearing capacity [8], deflection [9], deformation [10], load distribution [11], load input [12], influence line [13], influence surface [14], modal parameters, etc. [15,16], can thus be reflected by the structural displacement. Among them, structural displacement monitoring based on computer vision has attracted more and more attention because it has many advantages, e.g., non-contact, high accuracy, time and cost saving, multi-point monitoring, etc. [17]. Computer vision monitoring methods of structural displacement have been applied to many tasks of bridge health monitoring, Yoon et al. [18] used a UAV(Unmanned Aerial Vehicles) to carry a 4K camera to monitor the displacement of a steel truss bridge, and obtained the absolute displacement of the structure without the influence of UAV movement. Ye et al. [19] used a computer-controlled programmable industrial camera to monitor the behavior of an arch bridge under vehicle load, obtaining the influence line of its structural displacement, and realizing the real-time online displacement monitoring of multiple bridges. Tian et al. [20] combined the acceleration sensor and visual displacement measurement method to carry out an impact test of a structure and construct the frequency response function of the structure, to realize the estimation of the structure’s mode, mode shape, damping, and modal scale factor, and to realize the impact displacement monitoring of the pedestrian bridge. Besides the bridge health monitoring, many other engineering applications also use this method to monitor and identify structural displacements. For example, Chang et al. used structural displacement monitoring, feature extraction, and the support vector machine of computer vision to form vibration monitoring systems for the on-site diagnosis and performance evaluation of industrial motors and carried out preventive maintenance experiments [21]. Liu studied a track displacement monitoring system in which a fixed camera at the trackside was used for imaging and then the actual displacement of the track was calculated through a digital image processing algorithm, which realized an accurate non-contact measurement of track displacement [22]. Based on the above studies using computer vision to monitor structural displacement in different engineering applications, it can be seen that it is effective, convenient, and accurate to monitor structural states by detecting displacement signals.

Based on the existing bearing monitoring are some shortcomings of traditional methods (including installing additional sensing devices on the bogie which increases manufacturing cost; trackside monitoring is susceptible to ambient noise, etc.). In this article, a displacement monitoring method based on computer vision to monitor the vertical displacement of the axle box bearing of the rail vehicles under simulated real working conditions in proposed, in order to realize the non-contact, high accuracy of axle box bearings of condition monitoring, the realization of ultimate axle box bearing fault diagnosis, and preventive maintenance. Firstly, a portable camera is used to image the platform and detect the displacement amplitude of the bolts, which is used to calculate the state of the bearing through the phase correlation method. Next, the displacement amplitudes of the bearing system in the vertical direction are derived by comparing the correlations of the image’s gray-level co-occurrence matrix (GLCM). Finally, for verification, the measured displacement is checked against the measurement from a laser displacement sensor. In the following sections, the proposed approach is used to monitor the displacement of several sets of the platform under different working conditions, and the experimental results are analyzed. Finally, the associated open research challenges are discussed.

## 2. Experimental Setup of Bearing Experimental Platform

The platform was built to simulate the real operation of the bearing, and at the same time, it can apply radial load (simulating the weight of the train) and stimulate the track irregularity to the tested bearing. The structure of the platform is shown in Figure 1. It mainly consists of three parts (shown in the dotted box in Figure 2): (1) the power input; (2) the excitation and expansion platform; (3) the whole bearing system. The platform adopts a horizontal structure, and the tested bearings are placed on both sides of the spindle. The spindle is connected to the motor through a universal joint coupling (simulating the speed of the train axle); the side view of the whole bearing system is shown in Figure 3. The bearing system is fixed on the excitation platform, and the end cover and the shell of bearing box are connected by eight bolts (bolts on which this article focuses) evenly distributed along the circumference. The radial load (simulated vehicle axle load) is applied to the bearing under test by two sets of vertically mounted springs. The radial load is loaded onto the two bearing under test. By applying vertical excitation to the test system (simulating the vertical irregularity of the actual line), the excitation platform is bolted to bearing bracket. The inertial force of the equivalent parts is reasonably simulated by the mass of the spindle and the bearing bracket. In this way, the bearing under the combined action of the radial load and excitation platform can simulate the working condition of the train operation so as to monitor the service state of the axle box bearing of a high-speed train. 

The electric machinery speed of the platform is 0~1460 r/min, the excitation frequency of the excitation platform is 0~50 Hz, and the vertical loading range of the tested bearing is 0~2000 kg. The parameters of the axle box bearing under test are shown in Table 1. To the platform can be added a temperature sensor, acoustic sensor, vibration sensor, etc., which have collected data including images, sound, temperature, and vibration.

## 3. Methodology

The general framework of the displacement monitoring system based on computer vision is shown in Figure 4. This section mainly describes the identification principles and methods of camera calibration, object tracking and displacement identification. The flow chart of displacement identification in this article is shown in Figure 5. Video input was collected by the portable camera (the video capture scheme is shown in Figure 6), and then captured the video into images according to the frame; all the images constitute the displacement image set. The displacement image of the first frame is marked as the sample image, and the coordinates of the three bolt points in the sample image are automatically located by the positioning algorithm. Then, images with the same coordinate position as the sample image are intercepted from the displacement image set and the sample image are calculated by the phase correlation method. Finally, the displacement amplitudes of the bearing system in vertical direction are derived by comparing the correlations of the image’s gray-level co-occurrence matrix (GLCM).

The portable camera was positioned on the central axis on the side of the platform; sample video images of the displacement of the platform for a short period of time are shown in Figure 7. The collected image contents include the bearing end cover and bolts, door frame, and expansion platform. The whole bearing system is fixed on the vibration platform, and the bearing end cover and bolts, door frame and expansion platform are fixed rigidly by bolts; therefore, the displacement amplitude of each place in the image collected in Figure 7 is the same. Therefore, in order to improve the calculation rate, and achieve the purpose of real-time monitoring and identification, the local images of the three bolts under the end cover of bearing box (as shown in the red rectangle in Figure 7) were used instead of the whole image to calculate the vertical displacement amplitudes of the three bolts over time. The displacement amplitudes in vertical direction of the three bolts over time were calculated individually. The displacement amplitudes of the bearing system in vertical direction were derived by comparing the correlations of the image’s gray-level co-occurrence matrix (GLCM).

### 3.1. Camera Calibration

For the displacement monitoring of the platform, the measurement task was limited to one-dimensional displacement; thus, this article adopted a simplified camera calibration method: the scale factor method [23].

As shown in Figure 8, when the camera optical axis is perpendicular to the structural plane, and the optical axis is in line with the normal of the structural plane, the calculation formula of scale factor *e* is:(1)$$e=\frac{D}{d}$$

Or
(2)$$e=\frac{Z}{f}d_{pixel}$$
where *D* is the size of the selected object in the structure plane, *d* is the corresponding number of pixels in the image plane, ƒ is the focal length of the lens, *Z* is the distance from the camera to the structure plane, and *d_pixel_* is the pixel size.

### 3.2. Object Tracking and Feature Extraction

The displacement calculation method used in this article is the phase correlation method; the method needs two Fourier transforms and one inverse Fourier transform. Therefore, the amount of calculation load increases with the input images and decreases with the operation speed. Therefore, it cannot meet the needs of real-time displacement monitoring. The simple compressed image size will lead to a decrease in the accuracy of the calculation of displacement, because the displacement amplitude of the captured image (as shown in Figure 9) is the same at every point. Observe the captured image, which contains the lower half of the end cover of the bearing box of the bearing system and the bolts above, part of the expansion platform and the door frame. Among them, the bolt features on the bearing end cover are obvious, the geometric features are stable, and the contour points are distributed uniformly along a central point, which reduces the calculation error of the power spectrum (Equation (12)) caused by the small and concentrated image pixel value gradient in the later phase correlation method to calculate the displacement. Therefore, the local image of the three bolts is selected for calculation of the phase correlation method. Figure 9 shows the displacement image of the platform under different ray directions due to different time periods (the difference can be seen clearly by the portion of the rectangular box in each drawing). Due to the influence of the change of image gray value caused by different ray direction on the positioning of bolts of end cover, this article compares two positioning methods: template matching positioning method and contour feature positioning method.

#### 3.2.1. Template Matching Positioning Method

The template matching and positioning method is shown in Figure 10 (taking one of the three bolts as an example). The main process is to intercept the grayscale image of the target bolt (the size of the captured image is $\sqrt{n}\times \sqrt{n}$) subset and the template in a part of the image in advance, and then traverse the matrix with the size of $\sqrt{n}\times \sqrt{n}$ in the grayscale pixel matrix of the image to be fixed. The two matrices are reduced to one dimension to perform correlation operation to complete the global search. The measurement index of the search is the normalized correlation coefficient r.
(3)$$r=\frac{n\sum_{i=1}^{n}x_{i}y_{i}-\sum_{i=1}^{n}x_{i}\cdot \sum_{i=1}^{n}y_{i}}{\sqrt{n\sum_{i=1}^{n}x_{i}^{2}-(\sum_{i=1}^{n}x_{i}{)}^{2}}\cdot \sqrt{n\sum_{i=1}^{n}y_{i}^{2}-(\sum_{i=1}^{n}y_{i}{)}^{2}}}$$
where $x_{i}$ represents the element corresponding to the reduction in one dimension of the template matrix, and $y_{i}$ represents the element corresponding to the reduction in one dimension of the truncated matrix with the size of $\sqrt{n}\times \sqrt{n}$ in the grayscale pixel matrix of the image to be determined.

The correlation coefficient r ranges from 0 to 1, and the peak position in the correlation coefficient map in Figure 10 is the matching position. In Figure 10, the abscissa corresponds to the abscissa of the image, the ordinate corresponds to the ordinate of the image, and the vertical coordinate is the correlation coefficient. The horizontal and vertical coordinates are in pixels, and the vertical coordinates are unified dimensions. Figure 11 shows the effect picture of the image taken under different illumination conditions in Figure 9. When the size of the image to be positioned is input as 960 × 544, the size of the template is set as 51 × 51 and the number of templates is 1; the calculation time is shown in Table 2 (the computer CPU used for calculation was Inter(R) Core(TM) I7-4790 CPU @ 3.60 GHz).

According to the positioning effect image in Figure 11 and the calculation time in Table 1, it can be seen that the template matching method has a good positioning effect and high robustness to the illumination conditions of the image. However, this method needs to traverse the entire image removal matrix and calculate multiple correlation coefficients, so the calculation time is relatively long, and the average positioning time is 304.5 seconds per piece, which is difficult to meet the real-time displacement identification requirements of the platform. 

#### 3.2.2. Contour Feature Positioning Method

The contour feature location method skips the gray pixel matrix of the image and directly uses the outstanding contour features in the image, establishing the geometric model through auxiliary graphics to complete the positioning. In order to more conveniently observe and select the contour features used in positioning, the image is first detected by canny edge, as shown in Figure 12. We can observe that the obvious and easy-to-detect contour features in the image are the two horizontal lines, *L*_1_ and *L*_2_ running through the whole image, and the contour features of these two lines are selected as the reference lines for the establishment of the geometric model. Observed that bolts *P*_1_ and *P*_3_ were symmetrically distributed about bolt *P*_2_, and bolt *P*_2_ was just on the center line *L*_3_ of the circular contour. *L*_1_, *L*_2_, *L*_3_, *P*_1_, *P*_2_, *P*_3_ and the circular contour were extracted from the edge image Figure 12 to establish the geometric model, as shown in Figure 13.

After the geometric model shown in Figure 13 is established, the positioning algorithm flow according to the established geometric model is shown in Figure 14. The specific steps are as follows.

Positioning reference line *L*_3_: canny edge detection for the image obtains the pixel matrix *A* of the image, as shown in Figure 12. The value of 255 in the matrix is the edge point, then the *x*-coordinate value of the reference line *L*_3_ is determined by Equation (4):
(4)$$X=\frac{X2-X1}{2}+X1$$
where *X*1 and *X*2 are the number of columns where the first edge point is located when traversing from the middle of the matrix to both sides in the first row of matrix A (as shown by the arrow in Figure 12).Positioning reference line *L*_1_/*L*_2_: Hough transform is adopted to detect the straight line, and the edge line of the excitation platform is positioned as the reference line, but there are two upper and lower edge lines *L*_1_ and *L*_2_, as shown in Figure 13. It can be observed that, when positioned as the lower edge line *L*_1_, the value of the pixel matrix of the grayscale image below the edge line is obviously smaller than that located at the upper edge line *L*_2_. According to Equation (5), whether the positioned edge line is the upper edge line *L*_1_ or the lower edge line *L*_2_ can be determined:
(5)$$\{\begin{array}{c} \begin{array}{ccc} L_{1} & & f(\frac{M_{2}+M_{1}}{2}+a,X)<b \end{array}\\ \begin{array}{ccc} L_{2} & & f(\frac{M_{2}+M_{1}}{2}+a,X)\geq b \end{array} \end{array}$$
where *M*_1_ and *M*_2_ are the vertical coordinates of the two endpoints of the detected line, and *f(x, y)* is the gray value of the image at (*x*, *y*). *a* is the error value of the positioning edge line, and is the empirical parameter, *b* is the gray value below the edge line *L*_2_, and is an empirical parameter. In this article, *a* = 6, *b* = 40. 

When the positioning reference line is determined to be *L_1_* or *L*_2_, the vertical distance between the reference line and the bolt *P*_2_ is *d_1_* or *d_2_*: (6)$$d_{1}=H*k_{1}$$
(7)$$d_{2}=H*k_{2}$$
where *H* is the height of the image; *H* = 544 in this article. *k*_1_ and *k*_2_ are empirical parameters, *k*_1_ = 0.1; *k*_2_ = 0.25.

Positioning bolts: after positioning the reference line, and according to the geometric relationship shown in Figure 13, the locations of the center points of the bolt are at the positions *P*_1_(*x*1, *y*1), *P*_2_(*x*2, *y*2), *P*_3_(*x*3, *y*3). When the positioning edge line is *L*_1_:
(8)$$\{\begin{array}{l} \begin{array}{l} \begin{array}{l} x1=X*(1-k_{3})\\ x2=X \end{array}\\ x3=X*(1+k_{3})\\ y2=\frac{M_{2}+M_{1}}{2}-d_{1} \end{array}\\ y1=y3=\frac{M_{2}+M_{1}}{2}-d_{1}-k_{4}*H \end{array}$$


When the positioning edge line is *L*_2_:(9)$$\{\begin{array}{l} \begin{array}{l} \begin{array}{l} x1=X*(1-k_{3})\\ x2=X \end{array}\\ x3=X*(1+k_{3})\\ y2=\frac{M_{2}+M_{1}}{2}-d_{2} \end{array}\\ y1=y3=\frac{M_{2}+M_{1}}{2}-d_{2}-k_{4}*H \end{array}$$
where *k_3_* and *k_4_* are empirical parameters.

After positioning to the center point of the bolts, the images of the bolts can be captured according to a certain size of the rectangular box. The positioning effect pictures are shown in Figure 15, and the calculation time is shown in Table 3 (the computer CPU used for calculation was Inter(R) Core(TM) I7-4790 CPU @ 3.60 GHz).

The red line in Figure 15a is the case of the reference line *L*_1_ and *L*_3_ of the positioning, and the case of the reference line *L*_2_ and *L*_3_ of the positioning is shown in Figure 15b–d. In the green rectangles are the locations of *P*_1_, *P*_2_ and *P*_3_ bolts. The method of using contour lines to establish reference lines to assist positioning has a good effect. Moreover, it can accurately identify and position objects under different lighting conditions and different surface textures, reflecting the robustness of the algorithm to surface texture changes and light intensity of objects. The average calculation speed is 0.02s to complete the positioning, which can meet the requirements of real-time monitoring and identification of the displacement of the platform.

Two different positioning methods of template matching and contour feature are compared. As can be seen from the positioning renderings in Figure 11 and Figure 15, the positioning effect of the two methods is good, and the robustness to ray intensity is very good, which can realize the automatic positioning requirements of bolts under different lighting conditions. According to the positioning times required by the two positioning methods in Table 2 and Table 3, the average positioning time of the template matching method is 304.5 s. The average of the contour feature method is 0.02 s, which greatly reduces the positioning time. Although the two positioning methods can achieve the same positioning effect, considering the requirements of real-time monitoring, this article adopted the second positioning method by contour feature for bolt positioning interception, and then carried out subsequent displacement calculation.

### 3.3. Displacement Calculation

In this article, the phase correlation method is used to calculate the displacement of the collected video images. The phase correlation algorithm [24] is a frequency-domain correlation algorithm based on Fourier power spectrum. This method only takes the phase information in the cross-power spectrum, which reduces the dependence on the image content. In addition, the obtained correlation peak is sharp and prominent, the detection range of displacement is large, and the matching accuracy is high. The image gray scale is less dependent and has a certain anti-interference ability. Assuming that *f*_2_(*x*, *y*) and *f*_1_(*x*, *y*) are two image signals, and *f*_2_(*x*, *y*) is obtained by *f*_1_(*x*, *y*) translation (*dx*, *dy*), which satisfies the following formula: (10)$$f_{2}(x,y)=f_{1}(x-dx,y-dy)$$

It is reflected in the frequency domain in the form of:(11)$$F_{2}(\mu ,\upsilon )=F_{1}(\mu ,\upsilon )*e^{-i*2\pi *(\mu *dx+\upsilon *dy)}$$

The cross-power spectrum of *f*_2_(*x*, *y*) and *f*_1_(*x*, *y*) can be obtained from the above formula:(12)$$H(\mu ,\upsilon )=\frac{F_{1}*F_{2}^{*}}{\left|A_{1}\right|*\left|A_{2}^{*}\right|}=e^{-i*2\pi *(\mu *dx+\upsilon *dy)}$$
where *F** is the conjugate of *F*.

The inverse Fourier transform of the cross-power spectrum can obtain a Dirac function (pulse function) and find the offset by finding the coordinates of the peak. However, this method can only obtain the displacement of the pixel level. Then, the peak position can be found according to the above, and a weighted mean of response size can be processed in *a* × *a* form centered on this position. The following formula can be applied to obtain the precision position at the sub-pixel level:(13)$$x=\frac{\sum_{a\times a}if(i,j)}{\sum_{a\times a}f(i,j)}$$
(14)$$y=\frac{\sum_{a\times a}jf(i,j)}{\sum_{a\times a}f(i,j)}$$

The final (*x*, *y*) is the subpixel displacement between the two images.

For the selection of the calculated displacement amplitudes, this article proposes a new method to convert the displacement amplitudes into images and calculate the correlation of the GLCM of the displacement amplitudes. The displacement amplitudes with the greatest correlation, namely the most periodic and obvious, are selected to represent the displacement of the platform.

In image processing, when not only the distribution of gray level but also the relative position of pixels in the image should be considered, the GLCM of the image is usually generated [25]. Let *Q* be an operator to define the relative position of two pixels, and consider an image f with L possible gray levels. Let *G* be a matrix, whose element *g_ij_* is the number of pixels with grays of *z_i_* and *z_j_* appearing in the position indicated by *Q* in f, where 1 ≤ *i*, *j* ≤ *L*. The matrix formed in this way is called GLCM. Figure 16 shows an example of how the GLCM is constructed using *L* = 8 and the position operator *Q* defined by “one pixel to its right”. The array on the left is the small image in consideration, and the array on the right is the matrix *G*.

Element (1,1) of the GLCM *G* is 1, because in f, the pixel with value of 1 to the right of a pixel with value of 1 appears only once. Similarly, the element of *G* (6,2) is 3, because in f, the right value of a pixel with value of 6 appears three times with value of 2, and the possible gray level of the image determines the size of the matrix *G.* The total number of pixel pairs *n* that satisfy *Q* is equal to the sum of the elements of *G*. As a result: (15)$$p_{ij}=g_{ij}/n$$

Correlation is a descriptor of the characteristics of the GLCM and is a measure of how closely a pixel is related to its neighbors on the entire image. The range is [1,−1], which corresponds to perfect positive correlation and perfect negative correlation. The correlation is calculated as follows: (16)$$\sum_{i=1}^{K}\sum_{j=1}^{K}\frac{(i-m_{r})(j-m_{c})p_{ij}}{\sigma_{r}\sigma_{c}}$$

In Equation (16), the quantity used is defined as follows: (17)$$m_{r}=\sum_{i=1}^{K}i\sum_{j=1}^{K}p_{ij}$$
(18)$$m_{c}=\sum_{j=1}^{K}j\sum_{i=1}^{K}p_{ij}$$
(19)$$\sigma_{r}{}^{2}=\sum_{i=1}^{K}{\left(i-m_{r}\right)}^{2}\sum_{j=1}^{K}p_{ij}$$
(20)$$\sigma_{c}{}^{2}=\sum_{j=1}^{K}{\left(j-m_{c}\right)}^{2}\sum_{i=1}^{K}p_{ij}$$
where *K* × *K* is the size of the GLCM.

## 4. Experimental Results and Analysis

### 4.1. Measured Results

Double row cylindrical roller bearings were used in the experiment; the displacement image of platform in stable state for 30 s was collected under the working conditions of rotation speed *n* = 0, static load *F* = 0 kg and excitation frequency *f* = 6 Hz. The displacement was calculated by using the phase correlation method of local images proposed in this article. The frame rate of the portable camera used in this test was 30 frames s^−1^; and the portable camera was placed on the central axis of the platform, and the camera calibration was carried out in accordance with Section 3.1. The computer CPU used for calculation was Inter(R) Core(TM) I7-4790 CPU @ 3.60 GHz. The sampling frequency in the experiment was more than twice that of the excitation frequency of the platform; therefore, the influence of temporal aliasing effect and rolling shutter effect on the image was not considered. According to the method proposed in Section 3.3., the vertical displacement amplitude of the input three bolts and the whole image were calculated, respectively. The four displacement amplitudes are shown in Figure 17, and the calculation times are shown in Table 4. The displacement amplitudes calculated from the three bolts were generated by the GLCM of position operator *Q* defined as “one pixel to the right of the bolt”. By calculating the correlation of GLCM, the amplitudes with the greatest correlation, namely, the strongest periodicity, were selected to represent the vertical displacement amplitudes of the platform. The correlation of the three bolts according to their respective GLCM is shown in Table 5.

By observing Table 4, it can be seen that the vertical displacement amplitudes calculated from the local images of input bolt *P*_3_ have the strongest periodicity; thus, the displacement amplitudes at bolt *P*_3_ are shown as the vertical displacement amplitudes of the platform.

### 4.2. Verification of Results

In order to verify the effectiveness and accuracy of the method in this article, the real displacement amplitude of the platform was collected by using a laser displacement sensor under the same working condition as in Section 4.1. Moreover, the comparison and verification were made from the peak mean value of the displacement amplitude (as shown in Table 6) and the spectrum diagram (as shown in Figure 18).

By comparing the spectrum diagram shown in Figure 18, it can be seen that the frequency of the vertical displacement amplitude of the platform calculated from the local image of the bolt *P*_3_ selected by the method described in Section 4.1. differs little from the actual measured value. From Table 5, the average peak of the three bolts’ displacement amplitudes and laser displacement sensor displacement amplitude were contrasted; bolt *P*_3_ is closer to the value measured by the sensor, and the error is 0.05 mm, which verifies that the selection of displacement amplitude based on the correlation of the GLCM is effective.

In this article, local (bolt) images were input to replace the whole image for displacement calculation, and GLCM correlation of different displacement amplitudes was calculated by converting the displacement amplitudes into images, and the method of final calculation result of displacement amplitudes with the maximum correlation was selected. By observing Table 4 and Table 6, it can be seen that the calculation rate can be increased by 68% if the local image is input instead of the whole image, and the calculated displacement error can be guaranteed to be 0.05 mm.

### 4.3. Analysis of Displacement Amplitude under Different Working Conditions

The method verified in this article is used to calculate the displacement amplitude of the platform under different working conditions. The average values of the peak displacement amplitudes are shown in Table 7.

In Table 7, by comparing different working conditions, it is found that when static load and excitation frequency are unchanged, the vertical displacement amplitude of the platform tends to increase with the rotating speed. The vertical displacement amplitude of the platform tends to decrease with the static load when the rotational speed and excitation frequency remain unchanged.

## 5. Discussion

In this article, the vertical displacement of the platform was monitored by computer vision. The local image of the bolts on the bearing end cover were used instead of the whole image for calculation. Although the calculation speed can be increased by 68% with this method and an accuracy of 0.05 mm can be guaranteed, there are still several challenges that remain open for future investigation, and some critical challenges are discussed in detail below: (1) Limited frame rate and temporal aliasing effect: temporal aliasing is caused by the sampling rate (i.e., number of frames per second) of a scene being too low compared to the transformation speed of objects inside of the scene; this causes objects to appear to jump or appear at a location instead of smoothly moving, which can cause errors in the calculation of displacement. When the excitation frequency of the platform gradually increases and gradually exceeds the sampling rate of the camera, we cannot blindly adopt the camera with high frame rate. Therefore, the time aliasing effect needs to be studied and solved in the next stage of research. (2) Rolling shutter effect: most cameras use complementary metal oxide semiconductor (CMOS) sensors. The CMOS sensor uses a sequential readout, scanning each line exposed at different times to obtain the image, resulting in geometric distortion, especially when the relative velocity between the camera and the object is high. As the excitation frequency of the platform increases, in order to reduce the error, additional research is needed to eliminate the shutter curtain effect in the case that the speed of the structure is large relative to the camera.

## 6. Conclusions and Future Work

This article proposed to use displacement signal to monitor the state of rail vehicle bearings. With the help of an experimental platform of bearing system, a method of displacement monitoring by computer vision detection was explored to identify the displacement. The vertical displacement of all the components in the whole bearing system was the same, to improve the calculation rate and meet the purpose of real-time displacement monitoring; the bolts on the axle end cap were used for displacement identification. Two positioning methods were compared: the template matching method and contour feature method. Considering localization accuracy and localization efficiency, the localization method based on contour feature was chosen. The displacement amplitudes of the bearing system in the vertical direction were derived by comparing the correlations of the image’s gray-level co-occurrence matrix (GLCM). The measured displacement of the laser displacement sensor was compared with the calculated displacement in the frequency domain and the mean value of the peak displacement to verify the accuracy of the proposed method. 

The following conclusions and findings are drawn: Contour feature to locate the bolt is much faster than using the template matching method. The locating rate is 0.024 s/sheet on average, and the bolt is more robust to illumination conditions;By replacing the whole image with a local image, the phase correlation method can improve the calculation rate by 68% with an accuracy of 0.05 mm;According to the correlation of GLCM of the displacement amplitude image, the displacement amplitude graph is the closest to the real value;The method of replacing the whole image with the local (bolt) images to calculate the displacement proposed in this article was used to calculate the displacement of six groups of bearing platforms under different test conditions;It was found that the vertical displacement amplitude of the bearing system increases with the increase in rotating speed and decreases with the increase in static load;It is feasible and effective to introduce displacement signals to monitor the state of bearings;At the same time, practical considerations and limitations of the proposed method were discussed, including the issues of the temporal aliasing effect, and rolling shutter effect. In addition, the following aspects are determined for future research: displacement amplitudes of the bearing system in the vertical direction should be derived by comparing more parameters of the image’s gray-level, rather than just by comparing co-occurrence matrix (GLCM); studying the influence of the acquisition distance on the accuracy of displacement calculation by setting the distance between different portable equipment placement points and the platform; diagnosing bearing faults by the displacement signals.

## Figures and Tables

**Figure 1 sensors-21-02100-f001:**
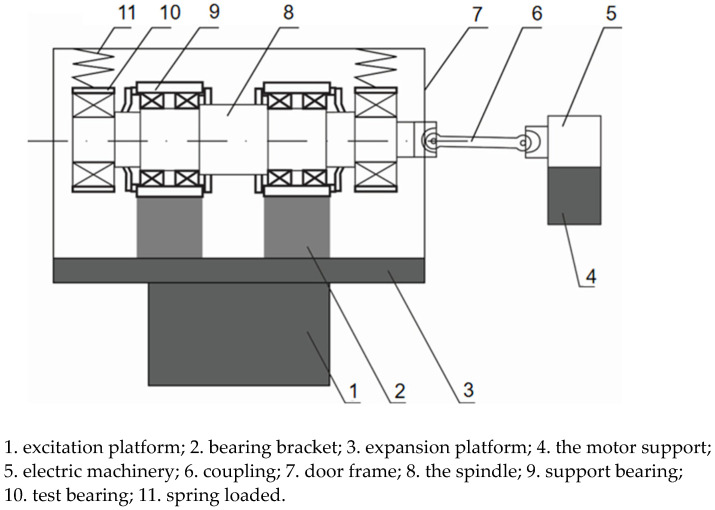
Structure of the bearing system experimental platform.

**Figure 2 sensors-21-02100-f002:**
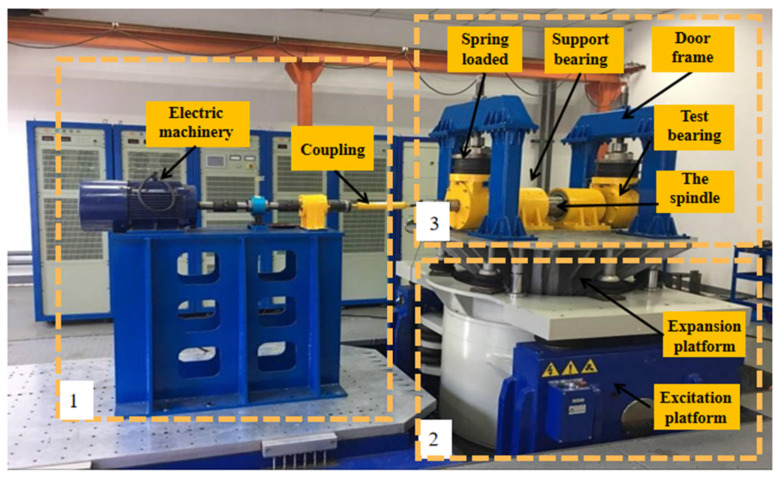
The design of the experimental platform.

**Figure 3 sensors-21-02100-f003:**
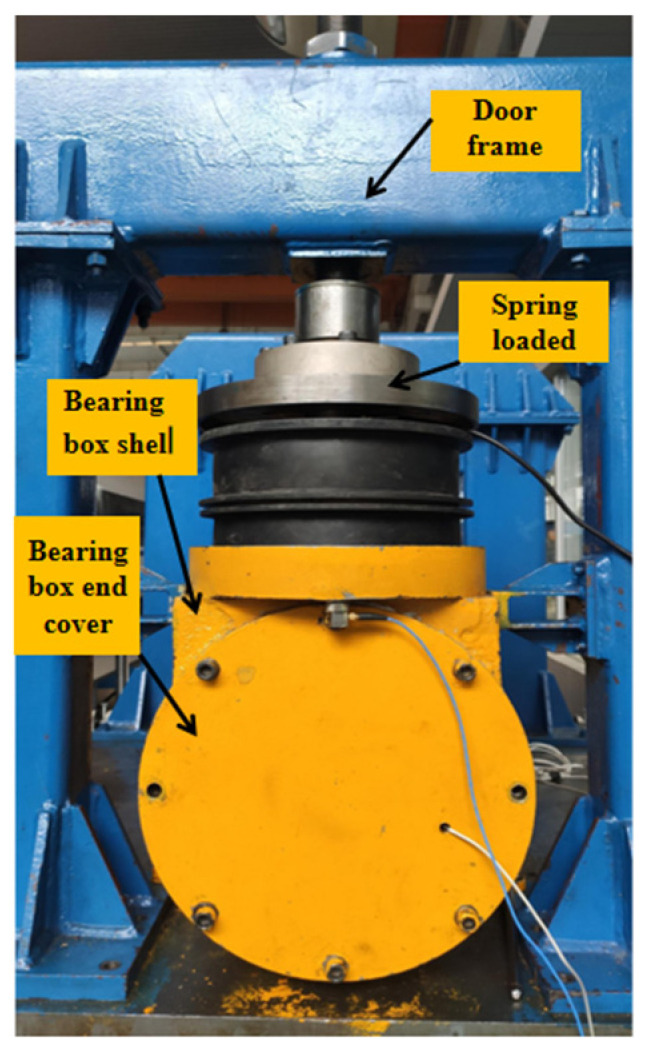
Side view of the bearing system.

**Figure 4 sensors-21-02100-f004:**
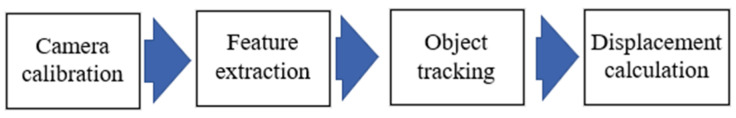
General framework of computer vision displacement monitoring.

**Figure 5 sensors-21-02100-f005:**
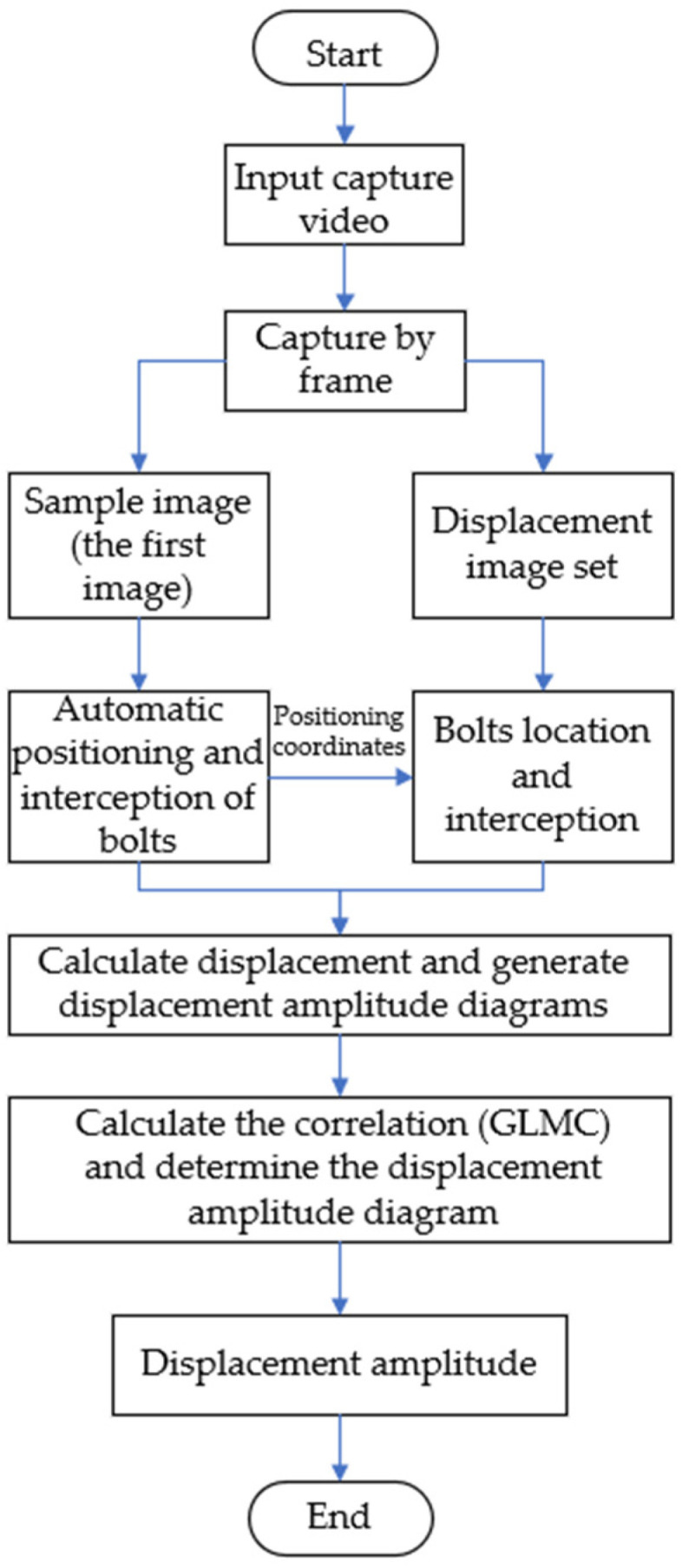
Flow chart of displacement identification.

**Figure 6 sensors-21-02100-f006:**
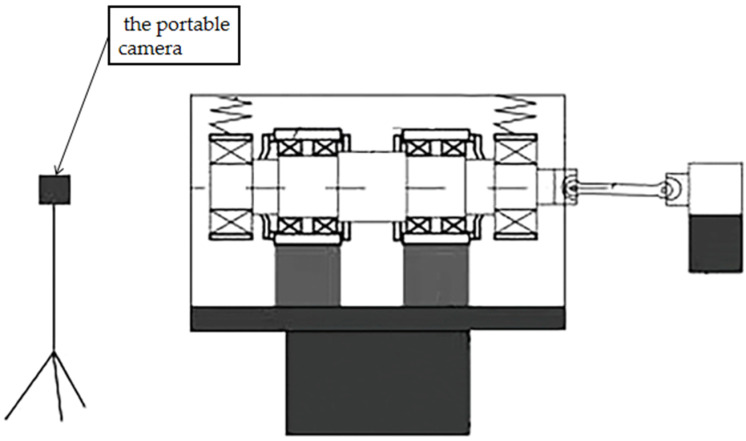
Displacement video capture scheme.

**Figure 7 sensors-21-02100-f007:**
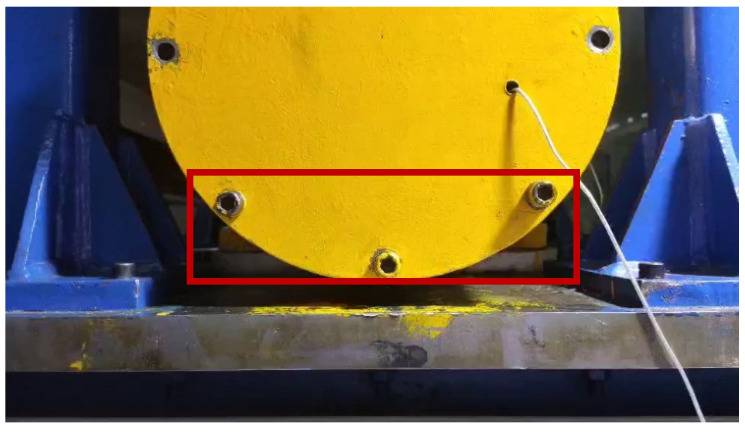
Image of bearing system experimental platform.

**Figure 8 sensors-21-02100-f008:**
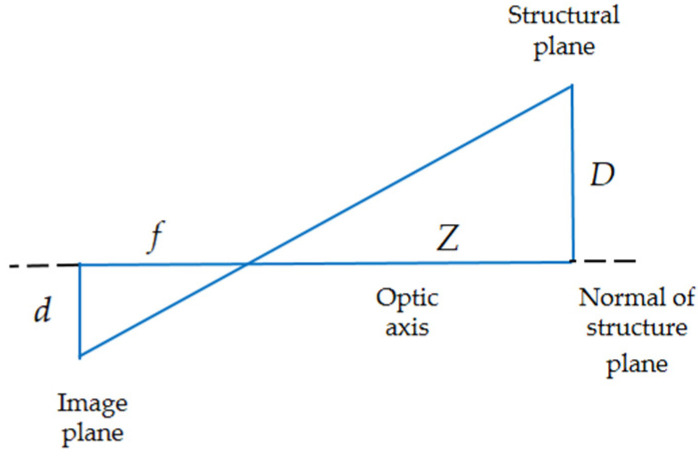
Calculation diagram of scale factor.

**Figure 9 sensors-21-02100-f009:**
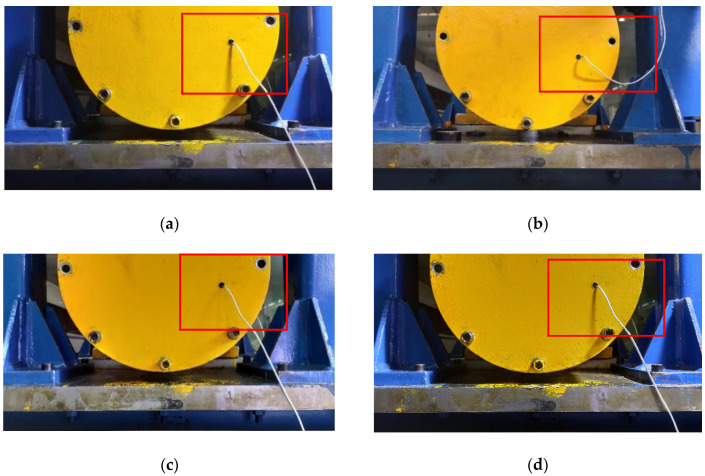
Images of the platform in different ray directions: (**a**) image under strong supplementary light source condition; (**b**) image under strong natural light conditions; (**c**) Images under low natural light conditions; (**d**) image under weak supplementary light source condition.

**Figure 10 sensors-21-02100-f010:**
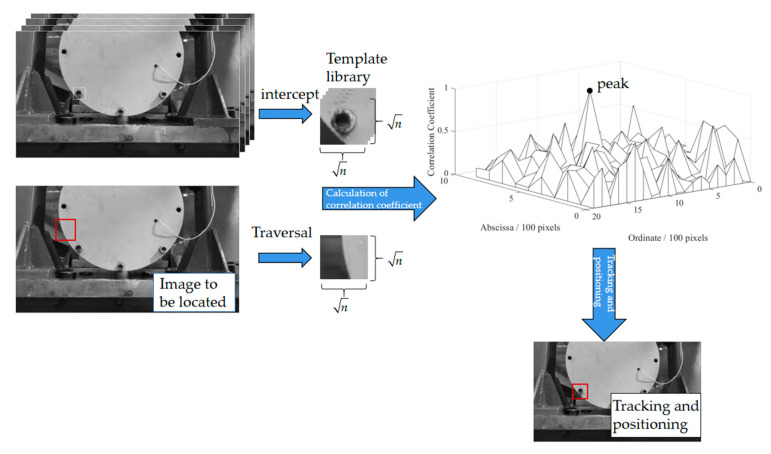
Template matching process.

**Figure 11 sensors-21-02100-f011:**
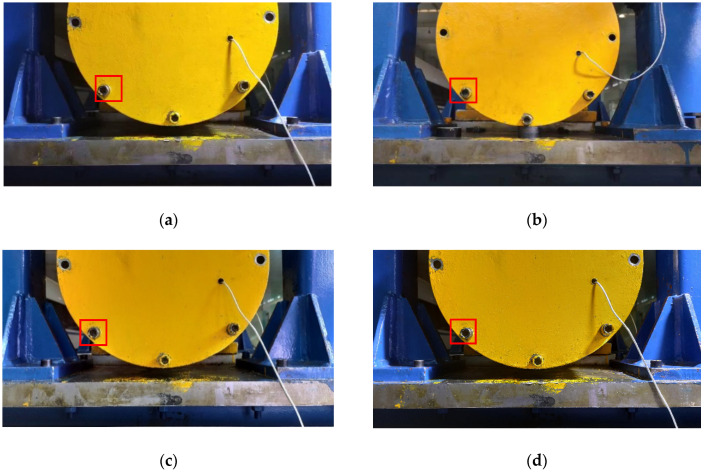
Template matching effect image under different illumination conditions (as Figure 9): (**a**) image under strong supplementary light source condition; (**b**) image under strong natural light conditions; (**c**) images under low natural light conditions; (**d**) image under weak supplementary light source condition.

**Figure 12 sensors-21-02100-f012:**
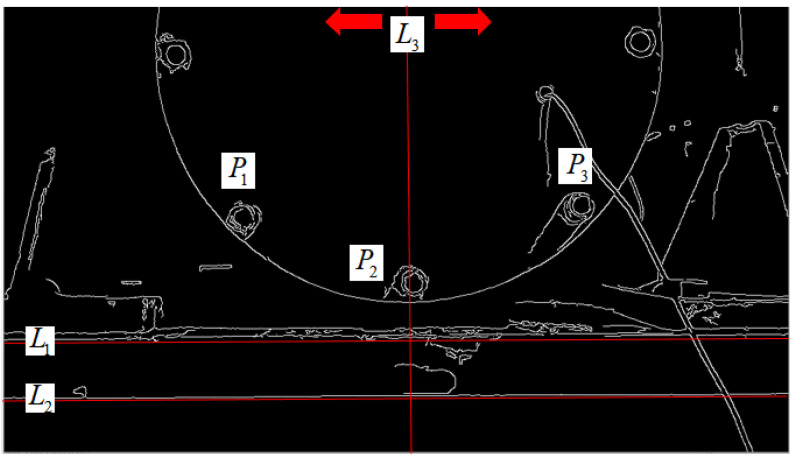
Effect of canny edge detection.

**Figure 13 sensors-21-02100-f013:**
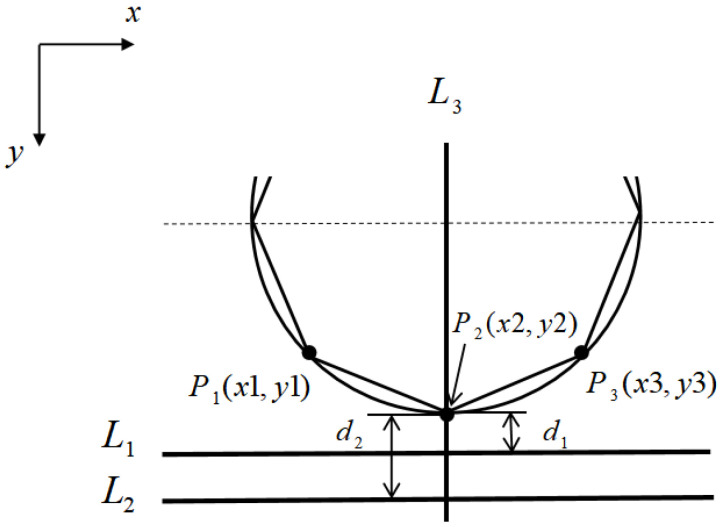
Simplified geometric model of end cover and bolt.

**Figure 14 sensors-21-02100-f014:**
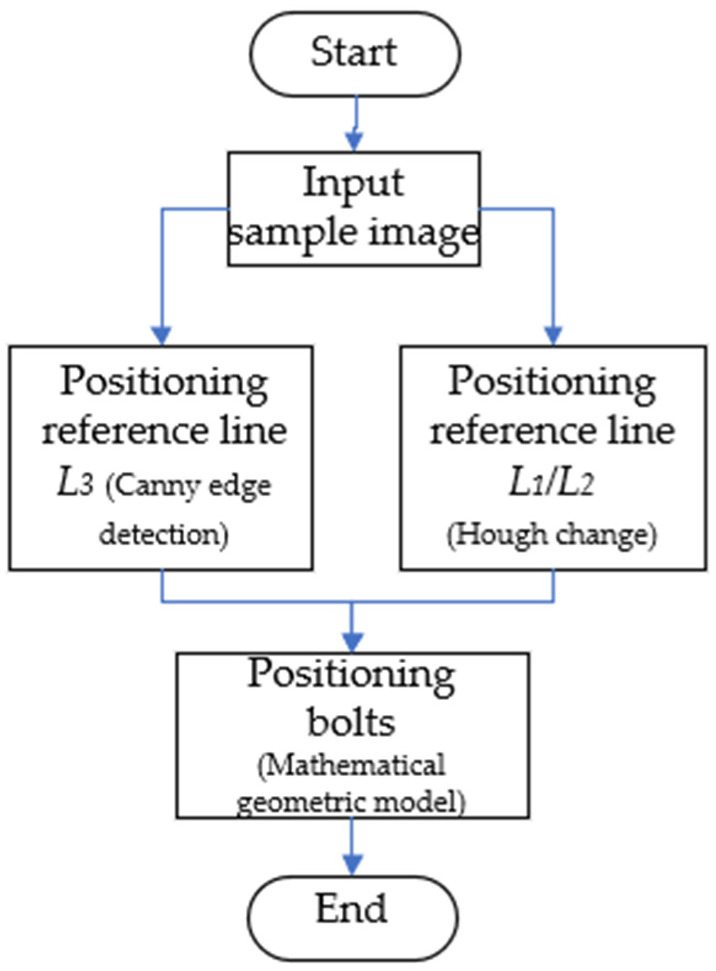
Bolt positioning flow chart.

**Figure 15 sensors-21-02100-f015:**
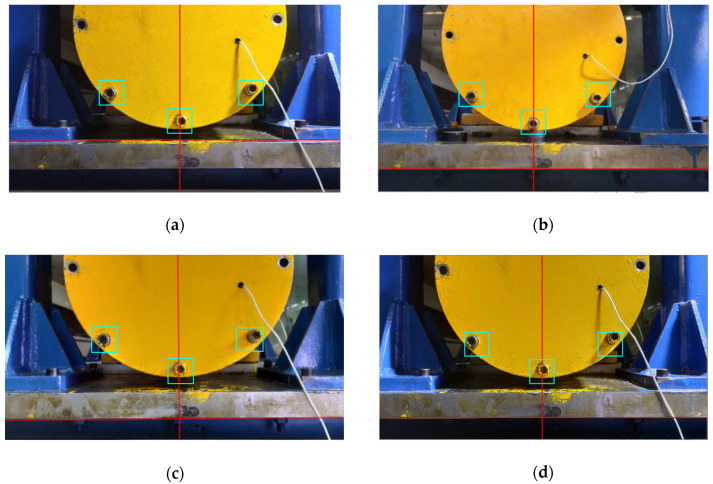
Contour feature positioning effect image under different illumination conditions (as Figure 9): (**a**) image under strong supplementary light source condition; (**b**) image under strong natural light conditions; (**c**) images under low natural light conditions; (**d**) image under weak supplementary light source condition.

**Figure 16 sensors-21-02100-f016:**
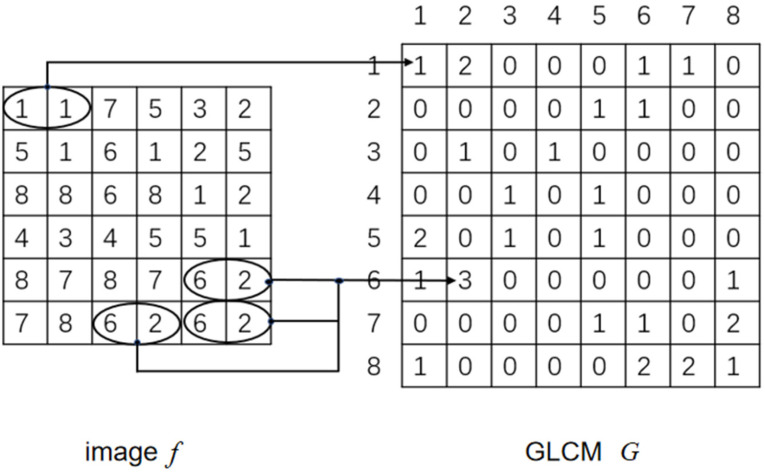
How to generate the gray-level co-occurrence matrix (GLCM).

**Figure 17 sensors-21-02100-f017:**
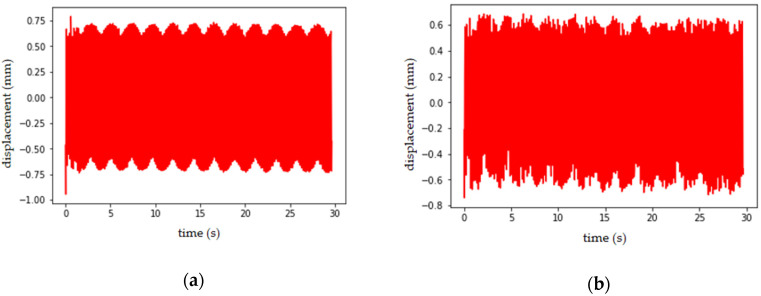
Vertical displacement amplitude diagram: (**a**) the vertical displacement amplitudes obtained when the input is a whole image; (**b**) the vertical displacement amplitudes obtained when the input is bolt *P*_1_; (**c**) the vertical displacement amplitudes obtained when the input is bolt *P*_2_; (**d**) the vertical displacement amplitudes obtained when the input is bolt *P*_3_.

**Figure 18 sensors-21-02100-f018:**
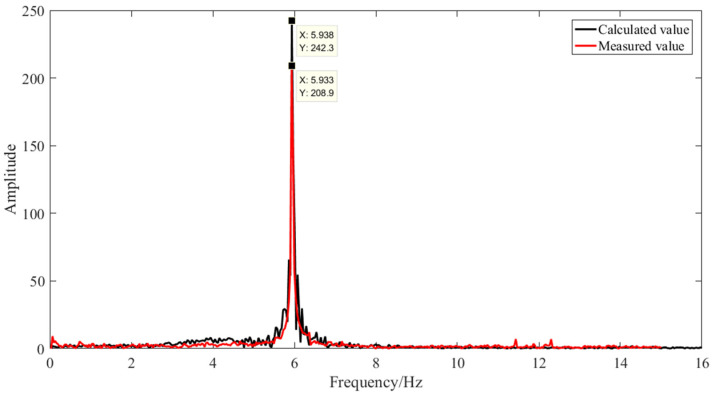
Spectrum comparison diagram.

**Table 1 sensors-21-02100-t001:** Axle box bearing dimension parameters.

Bearing Dimension Parameter	Parameter Value
bearing outside diameter/mm	240
bearing inner diameter/mm	130
bearing width/mm	180.5
roller diameter/mm	27
number of rollers	17 × 2

**Table 2 sensors-21-02100-t002:** Calculation times of template matching.

Image Number	Processing Time (s)
1	305.0
2	305.0
3	305.0
4	303.0
mean value	304.5

**Table 3 sensors-21-02100-t003:** Calculation times of contour feature positioning.

Image Number	Processing Time (s)
1	0.02
2	0.02
3	0.02
4	0.02
mean value	0.02

**Table 4 sensors-21-02100-t004:** Calculation times of input different images.

Input Image	Operating Time (s)
Whole image	24.199
Bolt $P_{1}$	6.895
Bolt $P_{2}$	7.252
Bolt $P_{3}$	7.000
Bolts $P_{1},\;\;P_{2},\;\;P_{3}$	7.221

**Table 5 sensors-21-02100-t005:** Correlation of the GLCM of different vertical displacement amplitudes.

Displacement Amplitude Diagram	Correlation (GLCM)
Bolt $P_{1}$	0.9098
Bolt $P_{2}$	0.9986
Bolt $P_{3}$	0.9991

**Table 6 sensors-21-02100-t006:** Peak mean values of displacement amplitudes of different displacement images.

Displacement Amplitude Diagram	Peak Mean (mm)
Bolt $P_{1}$	0.6041
Bolt $P_{2}$	0.6367
Bolt $P_{3}$	0.6465
Actual measurement	0.6973

**Table 7 sensors-21-02100-t007:** Mean peak values of displacement amplitudes under different working conditions.

Working Condition				Peak Mean (mm)
Number of Groups	Load (kg)	Rotation Speed (n/min)	Excitation (Hz)	
1	600	500	6	0.738
2	600	800	6	0.757
3	600	1100	6	0.855
4	1200	500	6	0.707
5	1200	800	6	0.718
6	1200	1100	6	0.737

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
