# Peer review of "Displacement Identification by Computer Vision for Condition Monitoring of Rail Vehicle Bearings"

_sensors, 2021, doi:10.3390/s21062100_

Round 1
Reviewer 1 Report
This paper considers the bearing system displacement measurement by using a camera. The positioning and displacement are investigated. The so called GLCM method is applied to the displacement calculation. The experimental result indicated that the displacement can be calculated with a sound precision.
However, this paper need a major revision. The main comments are listed in the following.
(1) At the beginning, the paper states the fault diagnosis issue of the railway vehicle bearing. Nevertheless, there is no any result related to the bearing fault diagnosis at all.
(2) Some figures in the paper can be combined together. for instance, Fig.1 and Fig.6
(3) In Fig. 11 Fig.9 and Fig.15, do we really need 8 subplots for each figure?
(4) The contour feature positioning section is hard to be followed, especially those formulas. Please state it clearly and easy to be followed.
(5) The time need to calculate and analysis is closely related to the computer and CPU (or GPU) you used. Please provide them in the revised version.
(6) The time need for the template matching is over 300s. That is related to the number of images in the library. Please provide the details.
The language need to be polished carefully.
Reviewer 2 Report
This paper presents a new method for monitoring the rail vehicle bearings by detecting the displacement using computer vision. While the paper introduce a novel method, the reviewer believes the issues below needs to be addressed in order to be published in this journal.
- Please enhance the literature review session. There are multiple studies on measuring structural displacement for health monitoring purpose. Please emphasize the novelty and the originality of this paper compared to the previous studies.
- Why did the authors use correlation based template matching methods? What are the pros and cons of using the method compared to optical flow based trackers?
- Have authors considered temporal aliasing effect? Since the frame rate of the camera used in this study is 30 fps, temporal aliasing can cause unexpected result especially above 15 Hz. Please read the articles below and comment on the temporal aliasing.
- Beberniss, T. J., & Ehrhardt, D. A. (2017). High-speed 3D digital image correlation vibration measurement: Recent advancements and noted limitations. Mechanical Systems and Signal Processing, 86, 35-48.
- Yoon, H., Shin, J., & Spencer Jr, B. F. (2018). Structural displacement measurement using an unmanned aerial system. Computer‐Aided Civil and Infrastructure Engineering, 33(3), 183-192.
- What type of image sensor did the authors used? If the authors have used CMOS sensor type, rolling shutter effect may caused error. Please discuss about this in the manuscript.
- Baker, S., Bennett, E., Kang, S. B., & Szeliski, R. (2010, June). Removing rolling shutter wobble. In 2010 IEEE Computer Society Conference on Computer Vision and Pattern Recognition (pp. 2392-2399). IEEE.
- Grundmann, M., Kwatra, V., Castro, D., & Essa, I. (2012, April). Calibration-free rolling shutter removal. In 2012 IEEE international conference on computational photography (ICCP) (pp. 1-8). IEEE.
- Yoon, H., Hoskere, V., Park, J. W., & Spencer, B. F. (2017). Cross-correlation-based structural system identification using unmanned aerial vehicles. Sensors, 17(9), 2075.
Reviewer 3 Report
Reviewer’s comments on a paper submitted to MDPI Sensors
“Displacement Identification by Computer Vision for Condition Monitoring of Rail Vehicle Bearing”
The authors performed a health condition monitoring based on computer vision for bearings of rail vehicles by detecting the displacement of bolts on end cap of bearing box. For fault diagnosis, vibration and temperature measured from the bogie and acoustic signals measured from trackside are often used. The health condition of bearing is reflected by the amplitude of the detected displacement by phase correlation method. They derive the displacement amplitudes of the bearing system by comparing the correlations of the image's gray-level co-occurrence matrix (GLCM).
The paper is well written and it makes good contribution to science and technology. The following comments are provided to authors to improve their work, before it can be accepted for publication in the journal:
[1] Many studies are referenced in the Introduction section without giving much information about their contributions. The authors should better outline the novelty their work and highlight the research gap which has been addressed in their study.
[2] At the end of section 1, add a paragraph to explain the organization of the study.
[3] In Section 2, an experiment test rig is set up to simulate the real operation of the bearings. Some further details are need regarding how this test rig was set up, what were the hardware/software requirements, what data is being collected, etc.
[4] Section 3 is too long. Some parts can be moved to Section 2 and/or Section 4. Please provide a reference for the framework. Where is this adopted from? Figure 5 also requires some further explanation.
[5] Equation (4) just simply represents the average of the number of columns X1 and X2? What is f in Equation (5)? Majority of equations are not provided with a reference. Please double check.
[6] Section 4 and Section 5 can be merged.
Reviewer 4 Report
The article presents the method of displacement identification for monitoring rail vehicle bearing by using computer vision. The article presents very interesting research and valuable for bearing system condition monitoring. The application of computer vision data and experimental platform are rightly chosen and sufficient. The test procedure is clear and justified. The authors of the article correctly described the study; it is important that the test procedure can be reproduced by other researchers. Below I have listed comments and suggestions for Authors.
- The section 5. Discussion does not refer to scientific discussion with similar test results. The discussion cannot focus only on the local aspect, which sums up the study but also presents the relationship between the results achieved and other results of this type of research. This lack of information should be supplemented by adding paragraphs in the parts of discussion.
- I recommend that you check the editing (punctuation, spaces, paragraph indents etc.).
Round 2
Reviewer 1 Report
The language can be further polished carefully.
Reviewer 3 Report
The authors have implemented almost all comments. The reviewer still believes section 5 (which has just 16 lines length) should be merged with Section 4. The English writing of the paper also must be improved.
Author Response
Please see the attachment

This manuscript is a resubmission of an earlier submission. The following is a list of the peer review reports and author responses from that submission.